# An Oral Fluorouracil Prodrug, Capecitabine, Mitigates a Gram-Positive Systemic Infection in Mice

Jack R. McLeod,[a*] [ID]Pamela A. Harvey,[a] [ID]Corrella S. Detweiler[a]

[a]Department of Molecular Cellular and Developmental Biology, University of Colorado, Boulder, Colorado, USA

Pamela A. Harvey and Corrella S. Detweiler contributed equally. Author order was determined by the lab in which the project originated, that of the last author.

**ABSTRACT** New classes of antibiotics are needed to fight bacterial infections, and repurposing existing drugs as antibiotics may enable rapid deployment of new treatments. Screens for antibacterials have been traditionally performed in standard laboratory media, but bacterial pathogens experience very different environmental conditions during infection, including nutrient limitation. To introduce the next generation of researchers to modern drug discovery methods, we developed a course-based undergraduate research experience (CURE) in which undergraduate students screened a library of FDA-approved drugs for their ability, in a nutrient-poor medium, to prevent the growth of the human Gram-negative bacterial pathogen *Salmonella enterica* serovar Typhimurium. The nine drugs identified all disrupt DNA metabolism in bacteria and eukaryotes. One of the hit compounds, capecitabine, is a well-tolerated oncology drug that is administered orally, a preferred treatment route. We demonstrated that capecitabine is more effective at inhibiting *S.* Typhimurium growth in nutrient-limited than in standard rich microbiological broth, an explanation for why the antibiotic activity of this compound has not been previously recognized. Capecitabine is enzymatically converted to the active pyrimidine analogue, fluorouracil (5-FU), and Gram-positive bacteria, including *Staphylococcus aureus*, are significantly more sensitive to 5-FU than Gram-negative bacteria. We therefore tested capecitabine for efficacy in a murine model of *S. aureus* peritonitis. Oral capecitabine administration reduced the colonization of tissues and increased animal survival in a dose-responsive manner. Since capecitabine is inexpensive, orally available, and relatively safe, it may have utility for treatment of intractable Gram-positive bacterial infections.

**IMPORTANCE** As bacterial infections become increasingly insensitive to antibiotics, whether established, off-patent drugs could treat infections becomes an important question. At the same time, basic research has revealed that during infection, mammals starve pathogens for nutrients and, in response, bacteria dramatically alter their biology. Therefore, it may be fruitful to search for drugs that could be repurposed as antibiotics using bacteria grown with limited nutrients. This approach, executed with undergraduate student researchers, identified nine drugs known to interfere with the production and/or function of DNA. We further explored one of these drugs, capecitabine, a well-tolerated human oncology drug. Oral administration of capecitabine reduced infection with the human pathogen *Staphylococcus aureus* and increased survival in mice. These data suggest that capecitabine has potential as a therapy for patients with otherwise untreatable bacterial infections.

**KEYWORDS** antibiotics, capecitabine, course-based undergraduate research experience (CURE), 5-fluorouracil (5-FU), nucleoside analog, repurposing, *Salmonella enterica*, *Staphylococcus aureus*

Address correspondence to Pamela A. Harvey, pamela.harvey@colorado.edu, or Corrella S. Detweiler, detweile@colorado.edu.

* Present address: Jack R. McLeod, Dartmouth Geisel School of Medicine, Hanover, New Hampshire, USA.

Antimicrobial-resistant pathogens present a growing threat to human health. The rate of development of new antibiotics has not kept pace with the increasing incidence of resistance due to the difficulty of identifying new chemical classes of antimicrobials (1). Uncertainty

as to whether new antibiotics could be priced in a manner that would recover research costs and earn profits also hinders development (2). A rapid, cost-effective alternative to new therapies for untreatable infections is to identify off-patent, well-known drugs with the potential to be repurposed as antibiotics (3, 4).

Historically, antibiotics have been identified and evaluated by screening for chemicals that prevent bacterial growth in microbiological media that are nutrient rich and support many rounds of replication. This approach identified most of the clinical antibiotic classes currently in use (5, 6). It has the advantage of a large window of detection and is seemingly logical because bacteria replicate rapidly during the acute stages of infection. However, prior to rapid replication, bacterial pathogens need to overcome nutritional immunity, a host defense mechanism that restricts microbial access to metabolic building blocks (7). For example, growth of the human Gram-negative bacterial pathogen *S. enterica* serovar Typhimurium in mammalian cells is limited by access to amino acids (8, 9) and to carbohydrates or lipids (10, 11). Bacteria respond to nutritional restrictions by reprograming gene expression and activating diverse metabolic and catabolic pathways that enable virulence (12–14). Researchers have applied this information by screening for compounds that have antimicrobial activity in nutrient-poor media (15, 16).

We aimed to combine the scientific advances in antibiotic discovery described above with recent developments in the science, technology, engineering, and math (STEM) education field. Student involvement in research during undergraduate education promotes retention across STEM, reduces the time to graduation, and fosters identity as a scientist (17–19). To capture these benefits, course-based undergraduate research experiences (CUREs) engage students in authentic research (18). CUREs provide students with a meaningful connection to research faculty at their institution (20) and increase the participation of research faculty in teaching (18). For faculty, offering a CURE related to their research increases satisfaction with teaching and encourages sustained engagement (20). We worked with a CURE to identify off-patent drugs with potential utility for treating bacterial infections. The CURE screened a drug library for compounds that inhibit *S.* Typhimurium growth under nutrient-limiting conditions and identified nine drugs that prevent *S.* Typhimurium replication and are known to interfere with bacterial DNA metabolism. Two of the drugs, capecitabine and floxuridine, are prodrugs for fluorouracil (5-FU) that are used in patients to treat cancers (21, 22).

5-FU is a pyrimidine analog with multiple effects on eukaryotic and bacterial cells. It is enzymatically converted into a series of active metabolites that incorporate into DNA (dUTP) and cause DNA damage (23). 5-FU derivatives are also incorporated into mRNA (5-FUTP), causing miscoding and defective protein production (24). The combined effects of 5-FU inhibit bacterial growth but are not bactericidal (25). As with many drugs, 5-FU is considerably more potent against Gram-positive than Gram-negative bacteria (25–29). For example, a comparison of *Escherichia coli* and *Staphylococcus aureus* clinical isolates reported 5-FU minimum inhibitory concentrations that differ by 100-fold: 250 $\mu$g/ml versus 2 to 4 $\mu$g/ml, respectively (25). The heightened sensitivity of Gram-positive bacteria to 5-FU and other drugs (30, 31) is consistent with their lack of a protective outer membrane barrier (32, 33).

Floxuridine and one other 5-FU prodrug, 5-fluoro-2′-deoxyuridine (FdUrd), mitigate murine infection with *S. aureus*, a major human Gram-positive bacterial pathogen that causes skin and soft-tissue infections and forms biofilms on implants (34). Mice inoculated intraperitoneally (i.p.) with *S. aureus* have improved survival upon i.p. treatment with floxuridine starting 1 h after infection and continuing daily for 7 days (35). Similarly, neutropenic mice inoculated i.p. with *S. aureus* have improved survival upon i.p. treatment with FdUrd starting 1 h after infection and continuing daily for 3 days (36). Since floxuridine and FdUrd have antibacterial activity *in vivo* against *S. aureus* but are not, like capecitabine, orally available (37), we studied capecitabine further.

Capecitabine was developed in the late 1990s and is used to treat colon, breast, and gastric cancers (21, 37, 38). It is not by itself considered antibacterial and requires three metabolic steps for conversion to 5-FU (29, 39). To validate the identification of

capecitabine as an inhibitor of *S.* Typhimurium growth, we compared growth in the presence of capecitabine in the minimal medium versus a rich medium. Since *S. aureus* is more sensitive to capecitabine and other 5-FU prodrugs than *S.* Typhimurium (25–28, 35, 36, 40), we determined whether capecitabine delivered orally mitigates *S. aureus* systemic infection in mice.

## RESULTS

**Screen of drug library for inhibition of bacterial replication in a nutrient-poor medium.** To identify drugs with antibacterial activity under nutrient-limiting conditions, we worked with students in a CURE to screen a library of 129 drugs approved by the United States Food and Drug Administration (FDA), the Oncology Drug Set VII from the National Cancer Institute. We used a virulent strain of *S.* Typhimurium (SL1344) grown overnight in a rich medium, lysogeny broth (LB) (41, 42). Cultures were diluted in M9 minimal medium supplemented with 2% glucose. Students added bacteria into each well of a 96-well plate, followed by the addition of library compounds to a final concentration of 10 $\mu$M. Control samples included vehicle and ampicillin. Plates were incubated at 37°C without shaking for 24 h, and the optical density at 620 nm ($OD_{620}$) was measured to estimate bacterial growth. Vehicle-treated bacteria reached an average $OD_{620}$ of 0.192, whereas ampicillin-treated bacteria achieved an $OD_{620}$ of only 0.05 (Fig. 1). As expected, most library compounds had little or inconsistent effects on growth (see Table S1 in the supplemental material). Nine compounds consistently prevented bacterial growth, as defined by the cutoff of two standard deviations from the mean absorbance of the vehicle control. All nine hit compounds are drugs that interfere with DNA metabolism and have been used or explored as human cancer therapies (Table 1). Five are pyrimidines or pyrimidine prodrugs that inhibit DNA synthesis (azacytidine, capecitabine, decitabine, floxuridine, and gemcitabine). Plicamycin cross-links DNA, and carboplatin is a prodrug for a DNA cross-linker. Mitomycin alkylates DNA and also inhibits cellular thioredoxin reductase (43). Bleomycin is a mixture of glycopeptides that cleave DNA (44–46). Only one hit compound, capecitabine, is routinely administered to humans via an oral route (37). We therefore pursued this compound further.

**Nutrient limitation sensitizes *S.* Typhimurium to growth inhibition by capecitabine.** To establish whether we identified capecitabine because we screened in a nutrient-poor medium, we compared *S.* Typhimurium growth in the presence of capecitabine in M9 with 2% glucose to LB. *S.* Typhimurium was considerably more sensitive to capecitabine at 10 and 100 $\mu$M in M9 than in LB (Fig. 2A and B). We also compared the effect of a dose range of capecitabine on growth in M9 versus LB under conditions that as closely as possible mimicked the screen: resuspension of capecitabine in dimethyl sulfoxide (DMSO) instead of water, growth at 37°C without shaking, and reading of the plate on the spectrophotometer used for the screen, which has a 620-nm filter (Fig. 2C). Capecitabine was not effective at inhibiting *S.* Typhimurium growth in LB until concentrations of 100 $\mu$M were reached, whereas 12.5 $\mu$M capecitabine inhibited growth in M9. These observations suggest that experiments performed in rich media are unlikely to reveal the antibacterial activity of capecitabine. We conclude that under nutrient-limited conditions, *S.* Typhimurium is sensitive to growth inhibition by capecitabine and that the use of a nutrient-poor medium in the screen facilitated the identification of this drug.

**Capecitabine reduces *S. aureus* colonization of tissues in mice.** 5-FU is more potent against Gram-positive than Gram-negative bacteria (25–29), and two 5-FU prodrugs improve survival in *S. aureus*-infected mice (35, 36). Therefore, we tested capecitabine against *S. aureus* infection of mice. We used a clinical *S. aureus* human wound isolate (ATCC 6538) that is virulent in mice (47–49). *S. aureus* was inoculated intravenously into BALB/c mice at $5 \times 10^7$ CFU, as verified by plating on a selective medium (50–52). Capecitabine was delivered orally every 12 h over 5 days at 100 or 200 mg/kg of body weight. The dose of 400 mg/kg/day is the maximum tolerated by mice (53). We scored mice daily for health on the basis of coat, posture, and movement (54) (Fig. 3A). On day four, one mouse in the control group died, and its tissues were

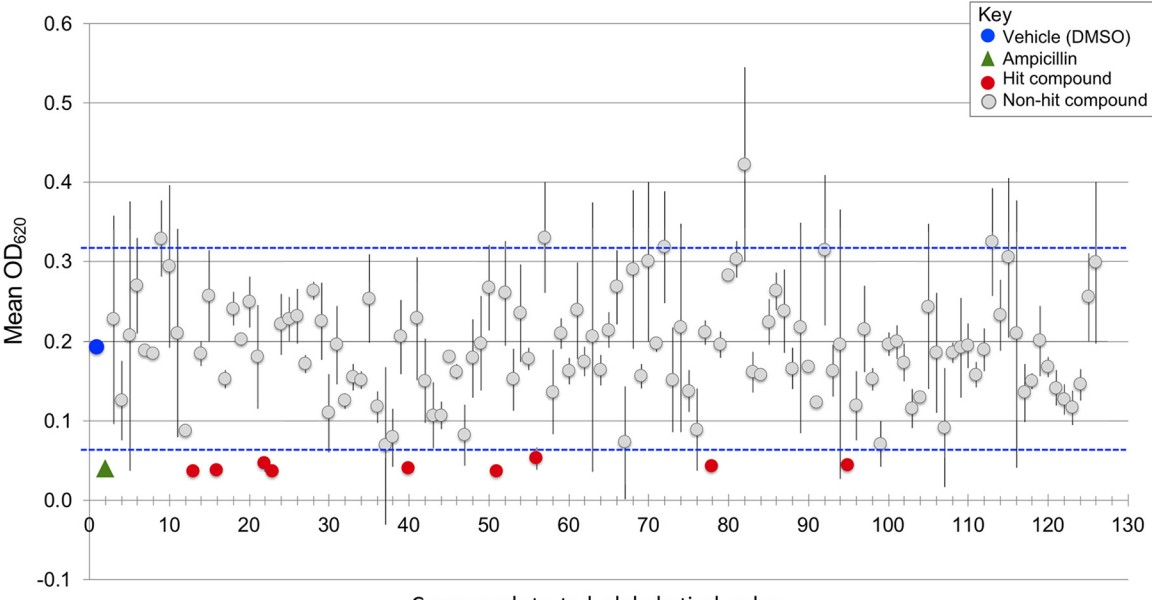

**FIG 1** Screen of a library of FDA-approved drugs by CURE students for the ability to inhibit *S.* Typhimurium growth in a nutrient-poor medium identified nine drugs. Shown are mean and SEM values at an $OD_{620}$ of 2 to 5 after 24 h of compound exposure in M9 minimal medium with 2% glucose at 37°C in static cultures. Vehicle (DMSO at 5%; blue circle) and ampicillin (50 $\mu$g/ml, 143 $\mu$M; green triangle) were used. Compounds are arranged alphabetically. The nine hits (red circles) are azacytidine, bleomycin, capecitabine, carboplatin, decitabine, floxuridine, gemcitabine, mitomycin, and plicamycin. Blue horizontal dashed lines represent two standard deviations from the means of the vehicle control, the cutoff for identifying a hit. The hits, DMSO, ampicillin, and several nonhit compounds have error bars that are too small to be visualized on the graph.

collected, homogenized, and plated to confirm colonization with *S. aureus*. After 5 days, remaining mice were euthanized. Tissues were harvested, weighed, homogenized, and plated for CFU enumeration. Mice treated with the higher dose of capecitabine had significantly better health scores and reduced bacterial tissue colonization compared to mock-treated mice (Fig. 3B). These findings indicate that capecitabine ameliorates acute *S. aureus* infection of mice.

## DISCUSSION

Here, we describe working with a CURE to screen for drugs that inhibit the growth of a Gram-negative bacterial pathogen, *S.* Typhimurium. We performed the screen in a nutrient-poor medium, whereas screens for new antibiotics are more typically performed in a richer medium, such as MHB, which was developed to optimize the growth of *Neisseria* species (6, 55). We identified multiple drugs known to interfere with DNA metabolism. Capecitabine was the only drug identified that is orally available and,

**TABLE 1** Hit compounds identified as inhibitors of *S.* Typhimurium growth in M9 medium

| Compound | Mean $OD_{600}$ | SEM | Mechanism of action; target, if known | Delivery route[a] |
|---|---|---|---|---|
| Vehicle (DMSO) | 0.192 | 0.003 | NA | NA |
| Ampicillin (control) | 0.040 | 0.000 | Inhibits cell wall synthesis | Oral |
| 13[b] Azacytidine | 0.036 | 0.000 | Inhibits DNA synthesis; pyrimidine, incorporates into DNA | i.v., s.c. |
| 16 Bleomycin | 0.037 | 0.001 | Inhibits DNA synthesis; unclear | i.m., i.p., i.v., s.c. |
| 22 Capecitabine | 0.046 | 0.010 | Inhibits DNA synthesis; prodrug for pyrimidine 5-FU, inhibitor of thymidine synthase | Oral, i.v. |
| 23 Carboplatin | 0.036 | 0.001 | Inhibits DNA synthesis; prodrug for cross-linker of DNA | i.v. |
| 40 Decitabine | 0.040 | 0.002 | Inhibits DNA synthesis; pyrimidine, incorporates into DNA | i.v. |
| 51 Floxuridine | 0.036 | 0.001 | Inhibits DNA synthesis; prodrug for pyrimidine 5-FU, inhibitor of thymidine synthase | i.v. |
| 56 Gemcitabine | 0.052 | 0.014 | Inhibits DNA synthesis; pyrimidine, incorporates into DNA | i.v. |
| 78 Mitomycin | 0.042 | 0.000 | Inhibits DNA synthesis; alkylates DNA, also other cellular targets | i.v. |
| 95 Plicamycin | 0.044 | 0.001 | Inhibits DNA synthesis; complexes with DNA and RNA | i.v. |

[a]i.m., intramuscular; i.p., intraperitoneal; i.v., intravenous; s.c., subcutaneous; NA, not applicable.
[b]Numbers correspond to the *x* axis in Fig. 1 and column A in Table S1.

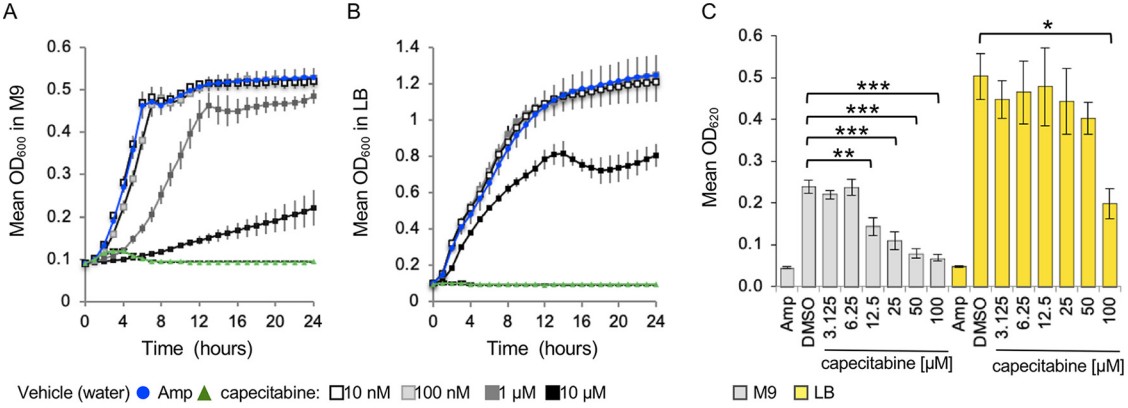

**FIG 2** Capecitabine is more effective at inhibiting the growth of *S.* Typhimurium in nutrient-poor than in rich media. (A and B) *S.* Typhimurium was grown in M9 minimal medium with 2% glucose (A) and LB (B) at 37°C with aeration in the presence of the capecitabine concentrations indicated. $OD_{600}$ was monitored hourly on a plate reader. Error bars represent SEM from four independent clones tested in triplicate. Vehicle (10% water) and ampicillin (50 μg/ml) were used. (C) Endpoint experiment performed under conditions mimicking the screen. *S.* Typhimurium was grown at 37°C standing, and $OD_{620}$ was monitored after 24 h. Vehicle (5% DMSO) and ampicillin (50 μg/ml) were used. Means with SEM from at least 6 biological replicates tested in triplicate are shown. ***, $P < 0.0001$; **, $P < 0.005$; *, $P < 0.05$, as established by analysis of variance with a Tukey-Kramer honestly significant difference.

therefore, of practical use for repurposing. We demonstrated that capecitabine is more potent against *S.* Typhimurium in minimal than in a standard rich medium, indicating that screening in nutrient-poor media facilitated the identification of capecitabine. These observations highlight that screens for new antibacterials performed under conditions that mimic infection identify different compounds than screens performed in traditional media (15, 56–58). They also underscore the value of working with students in a CURE format on screens of biological interest (59, 60).

Capecitabine is a 5-FU prodrug (21). Metabolites of 5-FU slow bacterial growth by interfering with DNA and RNA structure (23, 24). Two other 5-FU prodrugs, floxuridine and FdUrd, have been demonstrated to mitigate infection in mice inoculated with the

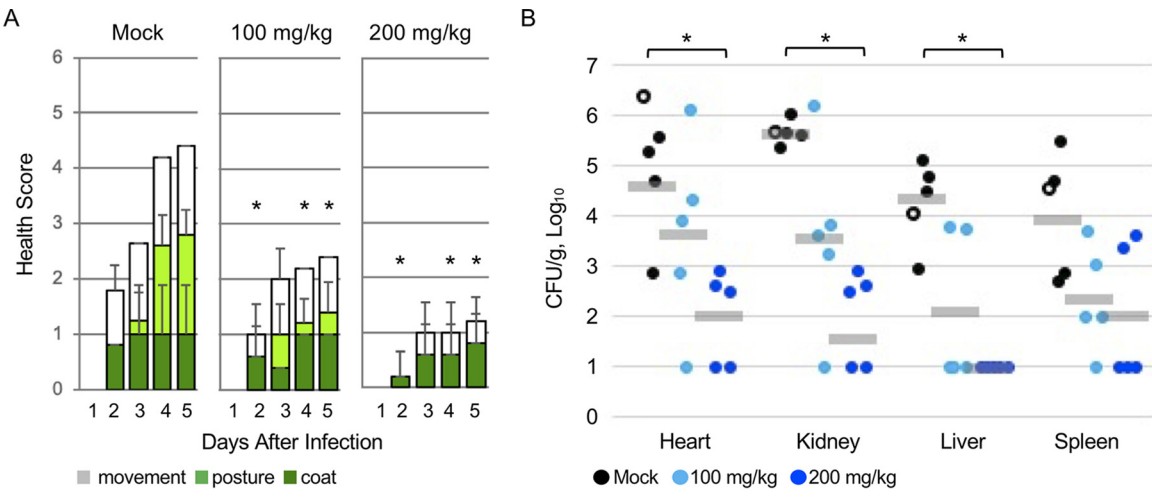

**FIG 3** Capecitabine improved health and reduced tissue colonization of *S. aureus*-infected mice. BALB/c female mice (8 to 9 weeks old) were infected intravenously with $5 \times 10^7$ CFU of *S. aureus*. Mice were treated orally with capecitabine at the dosage indicated or with sterile water (200 μl) every 12 h for 5 days. (A) Health was monitored every 24 h. Scores were summed to a maximum of six, and higher scores represent an increased incidence of morbid signs: movement (0 to 3), posturing (0 to 2), and coat condition (0 to 1). Means with standard deviations from 4 to 5 mice are shown. *, $P < 0.01$, relative to the corresponding mock-treated sample, as established by analysis of variance with a Tukey-Kramer honestly significant difference. (B) Tissue colonization. Each circle represents a tissue from a mouse, and 10 CFU/g was the limit of detection. Open circles show the number of CFU from the mock-treated mouse that died on day 4, although this mouse was excluded from the geometric mean (gray bars) and statistical analysis. *, $P < 0.05$, Mann-Whitney U test, two-tailed.

Gram-positive pathogen *S. aureus* (35, 36). 5-FU is approximately 100-fold more potent at inhibiting the growth of Gram-positive than Gram-negative bacteria (25–29). For these reasons, we tested capecitabine in mice inoculated with *S. aureus*, which demonstrated that capecitabine treatment reduced tissue colonization and increased murine survival.

The potential for *S. aureus* to evolve resistance to capecitabine is not practical to test in broth culture, because *S. aureus* is not growth inhibited by capecitabine at concentrations up to 200 $\mu$M (39). However, *S. aureus* 5-FU-resistant isolates evolved in MHB harbor mutations in genes consistent with the roles of 5-FU in disrupting DNA and RNA metabolism. The corresponding mutated proteins convert 5-FU into toxic pyrimidine analogs or increase concentrations of uracil, which competes with 5-FU analogs and thereby relieves toxicity (29). Resistant mutants grew similarly to the wild-type parent strain in broth, suggesting similar *in vitro* fitness levels (29). Whether the mutants are still able to cause infection in whole animals and remain resistant to 5-FU and/or capecitabine is unknown but seems unlikely given the centrality of the mutated genes to nucleotide metabolism and, therefore, bacterial replication.

Selection for bacterial mutants resistant to capecitabine in mice is similarly not feasible due to differences between capecitabine metabolism in primates and mice. Capecitabine is well absorbed in the murine and primate small intestine (61, 62). However, in murine intestinal epithelial cells, capecitabine is converted to 5-FU at rates at least 20 times higher than those in monkey or human intestinal cells (63). As a result, oral capecitabine causes severe nuclear degradation of intestinal crypt cells in mice at plasma 5′-DFUR concentrations that are benign in cynomolgus monkeys (63, 64). In primates, capecitabine travels from the intestine to hepatocytes, where the first two steps of catabolism occur. The third metabolic step, production of 5-FU, is catalyzed in many cell types (21, 37). Thus, the cytotoxicity of 5-FU is, in effect, distributed across the human body, mostly affecting rapidly dividing cells, such as in tumors. Mice are more than 6-fold less tolerant to oral capecitabine than humans: oncology patients typically receive 2,500 mg/kg/day of oral capecitabine, whereas mice tolerate only 400 mg/kg/day (37, 53, 65, 66). Thus, it would not be possible to select for resistant *S. aureus* mutants in mice at capecitabine dosages equivalent to those received by patients.

In humans, capecitabine oral administration is well tolerated, and toxic effects do not generally become limiting for months (67). The most common toxicities caused by sustained treatment with capecitabine or 5-FU are cumulative and due to damage to rapidly dividing cells. Dose-limiting toxicities include diarrhea, cytopenias, hyperbilirubinemia, and cutaneous damage, such as hand-foot syndrome (palmar-plantar erythrodysesthesia) (67). Some patients experience central neurotoxicity and/or acute cardiotoxicity (68). However, over the short time frames consistent with antibiotic treatment, days to weeks, toxicity is rare (67).

Capecitabine tolerability and efficacy against cancer in humans is influenced by genetic polymorphisms in pyrimidine metabolic and degradative enzymes (37). Infection therapy would likely be subject to the same polymorphisms, but limitations on time and resources will likely make it difficult to *a priori* identify patients with reduced tolerability. However, there is an approved treatment for capecitabine overdose, uridine triacetate, which competes with the 5-FU derivative fluorouridine triphosphate for incorporation into RNA (69, 70). We conclude that due to its off-patent status, oral formulation, high tolerability, and evidence for efficacy against *S. aureus* infection in mice, capecitabine merits exploration as a potential antibiotic therapy or cotherapy for difficult-to-treat Gram-positive bacterial infections.

## MATERIALS AND METHODS

**Bacterial strains and media.** *S.* Typhimurium strain (SL1344) is a virulent calf isolate (71). SL1344 was grown in LB (41, 42) with 30 $\mu$g/ml streptomycin or in M9 medium (26.11 mM Na$_2$HPO$_4$·7H$_2$O, 22.04 mM KH$_2$PO$_4$, 18.70 mM NH$_4$Cl, 8.56 mM NaCl, 0.1 mM CaCl$_2$, 63 $\mu$M MgSO$_4$) with 2% glucose and 0.04% histidine (SL1344 is a histidine auxotroph). *S. aureus* strain FDA209 (ATCC 6538) is a human wound isolate that was established as a standard strain for testing antiseptics by the Insecticide and Fungicide

Division of the U.S. Food and Drug Administration in 1938 (72, 73). Mannitol salt-selective agar (MSA) plates with phenol red were used to identify and enumerate *S. aureus* (74).

**Screen.** The primary screen was performed by undergraduate researchers within a CURE class (75). The library was the Approved Oncology Drug Set VII from the National Cancer Institute of the United States/National Institutes of Health Developmental Therapeutics Program (http://dtp.cancer.gov). We screened barcoded plates 4845 and 4846 (https://dtp.cancer.gov/organization/dscb/obtaining/available_plates.htm). The library was solubilized in DMSO (Sigma-Aldrich) at 10 mM and stored at −20°C.

*S.* Typhimurium cultures were grown overnight at 37°C in LB with aeration. Cultures were diluted 100-fold to approximately $5 \times 10^6$ CFU/ml in M9. To each 96-well round-bottomed test plate, 90 $\mu$l of diluted culture was added. Library compounds (10 mM) were diluted 1:100 into 50% DMSO and 10 $\mu$l was added to the plate, for final concentrations of 10 $\mu$M compound and 5% DMSO. Controls within each plate included DMSO and ampicillin (50 $\mu$g/ml; 143 $\mu$M). The concentration of ampicillin was chosen based on the need for a robust positive control given varied student pipetting techniques (59); we used the concentration of ampicillin routinely used to prevent the growth of *S.* Typhimurium in the laboratory (76). Plates were grown statically for 24 h at 37°C, and the $OD_{620}$ was the measured with an accuSkan FC microplate photometer (ThermoFisher Scientific, Waltham, MA). Biological replicates were independently performed by 2 to 5 student researchers. Outliers for individual compounds were determined using a Grubs test and eliminated. Compounds with values more than two standard deviations below the DMSO control were deemed hits.

**Growth curves and endpoint analyses.** *S.* Typhimurium cultures were grown overnight in LB at 37°C with aeration. Cultures were diluted to $\sim 5 \times 10^6$ bacteria/ml into M9 or LB. For the growth curve experiments, culture volumes of 180 $\mu$l were added to a flat-bottomed 96-well plate. Capecitabine was freshly diluted in water, and 20 $\mu$l volumes were added to reach the final concentrations indicated. Plates were grown for 24 h at 37°C with shaking, and the $OD_{600}$ was measured hourly in a Synergy 2 multi-mode or Eon microplate reader (BioTek, Winooski, VT). Mean $OD_{600}$ and SEM from four biological replicates tested in triplicate were calculated. For the endpoint analysis experiments, *S.* Typhimurium cultures were grown overnight at 37°C in LB with aeration. Cultures were diluted 100-fold to approximately $5 \times 10^6$ CFU/ml in M9 or in LB. Diluted culture (90 $\mu$l) was added to a 96-well round-bottomed plate. Capecitabine was freshly diluted in 50% DMSO and 10 $\mu$l was added to culture wells to reach the final concentrations indicated with 5% DMSO. Plates were grown statically for 24 h at 37°C, and the $OD_{620}$ was measured with an accuSkan FC microplate photometer (ThermoFisher Scientific, Waltham, MA).

**Ethics statement.** Animal work was carried out in accordance with the recommendations in the *Guide for the Care and Use of Laboratory Animals* of the National Institutes of Health (77). Euthanasia was carried out by carbon dioxide asphyxiation followed by cervical dislocation. All protocols were approved by Institutional Committee for Animal Care and Use (protocol number 2445) at the University of Colorado, Boulder, an Association for Assessment and Accreditation of Laboratory Animal Care (001743) accredited institution.

**Infection of mice.** Female BALB/c mice were purchased from Taconic Biosciences (Rensselaer, NY) and acclimated for 1 week. *S. aureus* were grown for 18 h overnight at 37°C in LB medium with aeration and diluted to $\sim 5 \times 10^7$ CFU/ml in phosphate-buffered saline (PBS) prior to inoculation. Mice that were 8 to 9 weeks old were weighed and then infected intravenously with $\sim 5 \times 10^7$ CFU of *S. aureus*, as determined by plating onto MSA and enumeration of CFU. Mice were treated orally by gavage with 200 mg/kg or 100 mg/kg capecitabine in 200 $\mu$l of water every 12 h for 5 days. Capecitabine was prepared 1 h prior to delivery by suspension in sterile water. Control animals were mock treated with water. Prior to dosing, visual health assessments were performed on the basis of movement (0 for normal, 1 for slow movement, 2 for movement after prodding, 3 for no movement with prodding), posture (0 for normal, 1 for hunched, 2 for prostrate), and coat health (0 for normal, 1 for a ruffled coat) (54). Scores were summed to a maximum of 6. Mice that succumbed to infection were scored as 6. Tissues were harvested, weighed, and homogenized in 3 ml of PBS. Diluted tissues were plated onto MSA and grown overnight at 37°C to enumerate CFU/ml and/or establish the presence of *S. aureus* in all tissues harvested. After 5 days of treatment, remaining mice were euthanized and tissues were similarly harvested and processed.

**Statistical analysis.** Statistical analysis was performed with GraphPad Prism 8 and with JMP version 15.

## SUPPLEMENTAL MATERIAL

Supplemental material is available online only.

**SUPPLEMENTAL FILE 1**, XLSX file, 0.9 MB.

## ACKNOWLEDGMENTS

We thank T. Su, L. Niswander, and M. Winey for support of the CURE laboratory, T. Mufford for expert assistance with animal infection experiments, and J. Dombach and C. Ewing for feedback on the manuscript.

The CURE was supported by a grant to Deborah Wuttke from the Howard Hughes Medical Institute and by the University of Colorado, Boulder, Department of Molecular, Cellular and Developmental Biology and the College of Arts and Sciences. Additional support was provided by NIH grant AI121365 (C.S.D.).

We declare no conflict of interest.

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
