## [Reviewer comments · Microbiology Spectrum]

Microbiology Spectrum

An oral fluorouracil prodrug, capecitabine, mitigates a Gram-positive systemic infection in mice

Jack McLeod, Pamela Harvey, and Corrella Detweiler

Corresponding Author(s): Corrella Detweiler, University of Colorado Boulder

Review Timeline:

Submission Date:	May 13, 2021
Editorial Decision:	May 18, 2021
Revision Received:	May 21, 2021
Accepted:	May 21, 2021

Editor: S. Wesley Long

Reviewer(s): The reviewers have opted to remain anonymous.

Transaction Report:

DOI: <https://doi.org/10.1128/Spectrum.00275-21>

May 18, 2021

Dr. Corrella S Detweiler
University of Colorado Boulder
MCD-Biology
347 UCB
Boulder, CO 80309

Re: Spectrum00275-21 (An oral fluorouracil prodrug, capecitabine, mitigates a Gram-positive systemic infection in mice)

Dear Dr. Corrella S Detweiler:

Thank you for submitting your manuscript to Microbiology Spectrum. As you will see your paper is very close to acceptance. Please modify the manuscript as follows:

While I appreciate your response to the reviewer comments, I agree with the reviewers that the stressed mice experiment is problematic, and would be very difficult to reproduce. As such, I would like to ask that you remove the data and text referring to the second mouse experiment from the manuscript, including Figure 4. I expect that you should be able to turn in the revised paper in less than 30 days, if not sooner.

When submitting the revised version of your paper, please provide (1) point-by-point responses to the issues I raised in your cover letter, and (2) a PDF file that indicates the changes from the original submission (by highlighting or underlining the changes) as file type "Marked Up Manuscript - For Review Only". Please use this link to submit your revised manuscript. Detailed information on submitting your revised paper are below.

Link Not Available

Sincerely,

S. Wesley Long

Preparing Revision Guidelines

- point-by-point responses to the issues I raised in your cover letter
- Upload a compare copy of the manuscript (without figures) as a "Marked-Up Manuscript" file.
- Each figure must be uploaded as a separate file, and any multipanel figures must be assembled into one file.
- Manuscript: A .DOC version of the revised manuscript
- Figures: Editable, high-resolution, individual figure files are required at revision, TIFF or EPS files are preferred

For complete guidelines on revision requirements, please see the Instructions to Authors at [link to page]. **Submissions of a paper that does not conform to Microbiology Spectrum guidelines will delay acceptance of your manuscript.**

Please return the manuscript within 60 days; if you cannot complete the modification within this time period, please contact me. If you do not wish to modify the manuscript and prefer to submit it to another journal, please notify me of your decision immediately so that the manuscript may be formally withdrawn from consideration by Microbiology Spectrum.

If you would like to submit an image for consideration as the Featured Image for an issue, please contact Spectrum staff.

May 21, 2021

Dr. Corrella S Detweiler
University of Colorado Boulder
MCD-Biology
347 UCB
Boulder, CO 80309

Re: Spectrum00275-21R1 (An oral fluorouracil prodrug, capecitabine, mitigates a Gram-positive systemic infection in mice)

Dear Dr. Corrella S Detweiler:

Thank you for removing the second experiment from the manuscript as I suggested. Your manuscript has been accepted, and I am forwarding it to the ASM Journals Department for publication. You will be notified when your proofs are ready to be viewed.

Sincerely,

S. Wesley Long
Editor, Microbiology Spectrum

Journals Department
Table S1: Accept